

# Evaluation of ultrasound-guided lateral thoracolumbar interfascial plane block for postoperative analgesia in lumbar spine fusion surgery: a prospective, randomized, and controlled clinical trial

Ke Chen[1,*], Lizhen Wang[2,*], Meng Ning[1], Lianjie Dou[3], Wei Li[4] and Yuanhai Li[1]

[1] Department of Anesthesiology, First Affiliated Hospital of Anhui Medical University, Hefei, Anhui, China

[2] Department of Anesthesiology, First Affiliated Hospital, University of Science and Technology of China, Hefei, China

[3] Department of Maternal, Child and Adolescent Health, School of Public Health,, Anhui Medical University, Hefei, China

[4] Department of Orthopedics (spine), First Affiliated Hospital of Anhui Medical University, Hefei, China

[*] These authors contributed equally to this work.

Corresponding author
Yuanhai Li, liyuanhai-1@163.com

## ABSTRACT

**Objective**. Ultrasound-guided lateral thoracolumbar interfascial plane block (US-TLIP block) is a novel regional technique for anesthesia or analgesia. However, there has been no prospective, randomized and controlled clinical trial investigating the perioperative analgesic effect of US-TLIP block on lumbar spinal fusion surgery. The aim of this study was to investigate the analgesic effect of bilateral single-shot US-TLIP in patients undergoing lumbar spinal fusion surgery.

**Methods**. A prospective and randomized comparative clinical study was conducted. A total of 60 patients (ASA classes: I–II), aged 21–74 years who were scheduled for lumbar spinal fusion surgery were randomized and divided into the TLIP group (Group T, $n = 30$) and control group (Group C, $n = 30$). The patients in Group T received preoperative bilateral single-shot US-TLIP with 30 ml of 0.375% ropivacaine at the third lumbar spine (L3) vertebral level, and the patients in Group C received an injection of 30 ml 0.9% saline through same technique. All patients received patient-controlled analgesia (PCA) after their operation. The frequency of PCA compressions and rescue analgesic administrations were recorded. Opioids (sufentanil and remifentanil), anesthetic consumption, the number of postoperative days spent in a hospital bed, overall hospital stay time and postoperative complications were recorded. The Visual Analogue Scale (VAS) and Bruggemann Comfort Scale (BCS) scores for pain and comfort assessment were recorded at 1, 12, 24, 36, and 48 hours postoperatively.

**Results**. Opioids and anesthetic consumption in the perioperative period decreased significantly in the TLIP group compared to the control group ($P < 0.05$). The VAS and BCS scores in the TLIP group were lower at 12, 24, and 36 hours postoperatively ($P < 0.05$). US-TLIP block has been shown to shorten postoperative hospital stays ($P < 0.05$). There was no significant difference in postoperative complications between the two groups.

**Conclusion**. Our study findings show that bilateral US-TLIP block exhibits significant analgesia and safety in patients undergoing lumbar spinal fusion surgery.

## INTRODUCTION

Posterior lumbar spinal fusion surgery, an effective method to limit the progression of deformity (*Weinstein et al., 2008*), has increased by 65% in the past 20 years (*Rushton et al., 2018*). This type of surgery brings about greater relief than classic conservative treatment (*Yoshihara, 2012*). However, patients receiving lumbar spinal fusion surgery often report reduced postoperative satisfaction and persisting postoperative pain, negatively affecting rapid postoperative recovery (*Gerbershagen et al., 2013*). Herein, pain management is a primary concern.

Clinically, there are many methods for pain control, including intravenous opioids, nonsteroidal anti-inflammatory agents, local anesthetic (LA) infiltration of incision sites, and regional analgesia. Each of these analgesic techniques possesses inherent advantages and disadvantages that restricts their universal applicability. For example, high doses of opioids effectively relieve pain in the perioperative period after spinal fusion. Unfortunately, the use of opioids is also associated with serious side effects, including addiction, respiratory depression, nausea and vomiting. These side effects often delay rapid patient recovery (*Manchikanti et al., 2017*; *Tobias, 2004*). Therefore, multimodal analgesia for proper control of pain appears to be the best strategy for pain management (*Bajwa & Haldar, 2015*).

The cornerstone of multimodal analgesia is regional analgesia (*Carli et al., 2011*; *Lenart et al., 2012*). Thoracolumbar interfascial plane block (TLIP block), a novel regional anesthesia technique, was first performed in 2015 (*Hand et al., 2015*). TLIP block effectively prevents the occurrence of pain via its action on the dorsal rami of spinal nerves (*Ahiskalioglu et al., 2018*; *Xu, 2017*; *Xu et al., 2019*). Gradually, it has become widely used in various surgical procedures. It has been reported that TLIP block (at the L3 vertebral level) provided an area of analgesia that covered the middle and had a predictable spread from L1 to S1 and from the left posterior axillary line to the right posterior axillary line in 10 participants (*Hand et al., 2015*). TLIP block can contribute significantly to a perioperative, multimodal, opioid-sparing analgesic regimen and enhance recovery time after lumbosacral spine surgery. However, there have been no reports of the application of TLIP block in randomized controlled trials of lumbosacral spine surgery. We conducted a randomized controlled trial to confirm whether or not the application of TLIP block could relieve pain after lumbosacral spine fusion surgery and significantly reduce perioperative opioid consumption, as part of multimodal analgesia for patients undergoing lumbosacral spine surgery.

## MATERIAL AND METHODS

The research study was performed at the First Affiliated Hospital of Anhui Medical University after approval from the local ethics committee (Ethics Committee of the First Affiliated Hospital of Anhui Medical University: Ethical Application Ref:2019-01-01MT).

We obtained written informed consent from all enrolled patients before surgery. The trial was registered at the Chinese Clinical Trial Registry (Registration No: ChiCTR1900022233). Patients, anesthesiologists, outcome assessors and data analysts were blinded to the study intervention.

A total of 102 patients were recruited to participate in the study. Patients were excluded from the study if they met any of the following criteria: abnormal liver and kidney function, psychiatric disorders or use of psychiatric medications, use of anticoagulants or corticosteroids, bleeding diathesis, or a known allergy to local anesthetics.

The patients were randomly divided into two groups using a computer program: control group (Group C, $n = 30$) and TLIP block group (Group T, $n = 30$). Two groups of patients were scheduled for lumber spine surgery. Before general anesthesia was administered, the patients in Group T were injected with ropivacaine through the application of bilateral TLIP block, while the patients in Group C were injected with saline using the same technique.

The TLIP block was performed after induction of general anesthesia, as described by Hand (*Hand et al., 2015*). This study was double-blind trails. To minimize bias error and unintentional physical cues, our patients were not aware of TLIP-block and the surgeon did not know what to inject. A high-frequency linear transducer was placed in the midline position at the third lumbar vertebra (L3), and 30 ml of 0.375% ropivacaine was injected bilaterally into the interfascial plane between the longissimus muscle (LF) and multifidus muscles (MFs) of patients in Group T. The corresponding procedure was performed using 30 ml of 0.9% saline for the patients in Group C. In our study, all TLIP block procedures were performed by the same anesthesiologists under the same medical conditions.

Standardized monitoring procedures were performed during anesthesia and surgery. Induction of general anesthesia was achieved by intravenously injecting propofol (1–2 mg/kg), sufentanil (0.3–0.5 µg/kg) and cisatracurium (0.2 mg/kg). After tracheal intubation, general anesthesia was maintained with propofol (4–10 mg/kg), remifentanil (0.25–4 µg/kg min) and cisatracurium (0.02–0.05 mg/kg h). By the end of skin closure, the anesthesiologists stopped the anesthetic agents and administered intravenous flurbiprofen (50 mg). After the operation, all patients were transferred to the postoperative recovery room and received PCA (sufentanil 4.5 µg/kg + 0.9% saline 150 ml, background dose 3 ml/h, self-control supplementary dose 3 ml, and locking time 10 min). BIS monitoring was performed in all patients, and BIS values were maintained at 40-60.

VAS and BCS scores were recorded at 1, 12, 24, 36, and 48 h postoperatively. The pain score was assessed using the VAS (choices ranging from 0 [no pain] to 10 (worst imaginable pain)). The postoperative comfort scale was assessed using BCS scores (0, continuous pain; 1, painless without movement, severe pain while breathing deeply or coughing; 2, painless without movement, mild pain while breathing deeply or coughing; 3, painless when breathing deeply; 4, painless when coughing). When the VAS score exceeded 5 at rest or with movement, patients were administered intravenous sufentanil (5 µg) one or more times. The frequency of PCA compressions and remedial analgesic administration was recorded. Postoperative complications were recorded by a nurse blinded to the study groups.

Our preliminary study showed that the average sufentanil consumption was 20 µg (±5) during the operation. A sample size of 22 subjects per group was calculated to be the
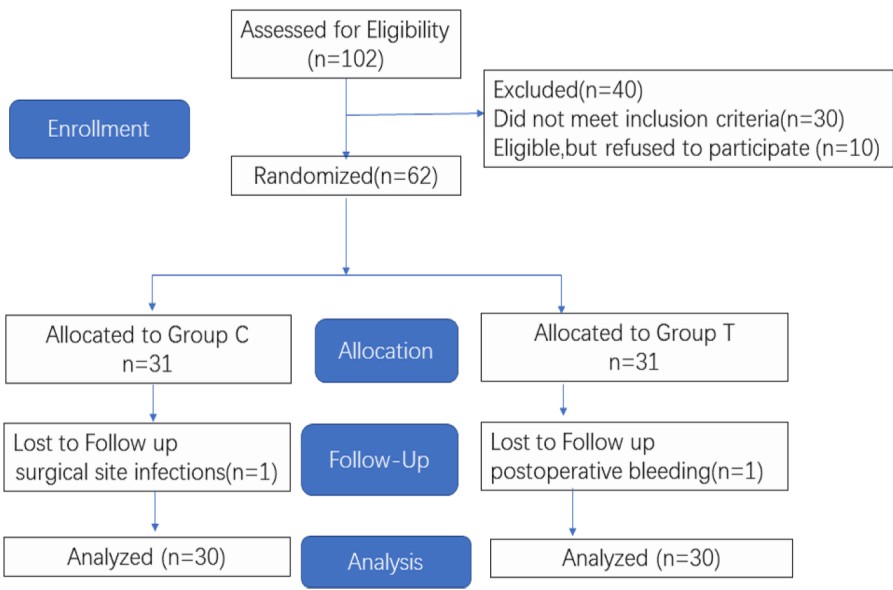

**Figure 1 CONSORT Flow Diagram.**

necessary number of participants needed to detect a 20% difference in perioperative time at 80% power with an error of 0.05. We included 30 patients in each group to make up for the possible withdrawal of patients.

All data analyses were performed using SPSS 23.0 software (SPSS, Chicago, IL, USA). The Kolmogorov–Smirnov test was used to detect the normality of data distribution. Continuous variables were expressed as the means ± standard deviations (M ± SD) and medians (25th–75th percentiles), while categorical variables were expressed as counts (percentages). Comparisons of normally distributed continuous variables between the two groups were performed using the Student's $t$-test, while nonnormally distributed continuous variables between the two groups were compared using the Mann-Whitney U test. Comparisons of categorical variables between the two groups were performed using the continuity corrected chi-square test. A $p$ value < 0.05 was considered statistically significant.

## RESULTS

Figure 1 shows the *CONSORT diagram* of enrollment for this study. In the study, 102 patients planning to undergo lumbar spinal fusion surgery were enrolled. A total of 40 patients were excluded, and a total of 62 patients were included in the randomized grouping process. Following randomization, one patient was excluded for a surgical site infection and another patient was excluded due to postoperative bleeding. Age, height, weight, ASA, body mass index (BMI), and surgical and anesthesia times showed no significant difference between the two groups (Table 1).

Figure 2 shows the distribution of the three muscles and the correct location of local anesthetic injection. An insulated echogenic needle (22-gauge,10 cm) was inserted in-plane

**Table 1  Baseline demographic data and clinical characteristics.**

| Characteristic variable | Group T | Group C | $\chi^2/t$ | $P$ value |
|---|---|---|---|---|
| Gender | | | $\chi^2 = 0.067$ | 0.796 |
| F | 14(23.33%) | 15(25.00%) | | |
| M | 16(26.67%) | 15(25.00%) | | |
| ASA status | | | $\chi^2 = 0.417$ | 0.519 |
| 1 | 5(8.33%) | 7(11.67%) | | |
| 2 | 25(41.67%) | 23(38.33%) | | |
| Age (y) | $58.65 \pm 8.51$ | $53.90 \pm 11.57$ | $t = -0.561$ | 0.577 |
| Weight (Kg) | $61.87 \pm 6.9$ | $63.20 \pm 11.03$ | $t = -1.403$ | 0.166 |
| Height (cm) | $165.07 \pm 7.64$ | $164.43 \pm 6.6$ | $t = 0.344$ | 0.732 |
| BMI (Kg/m$^2$) | $22.68 \pm 1.66$ | $23.25 \pm 3.0$ | $t = -0.910$ | 0.366 |
| Surgical time (min) | $173.17 \pm 38.18$ | $159.83 \pm 24.69$ | $t = 1.606$ | 0.115 |
| Anesthesia time (min) | $191.17 \pm 34.73$ | $188.33 \pm 26.7$ | $t = 0.354$ | 0.724 |

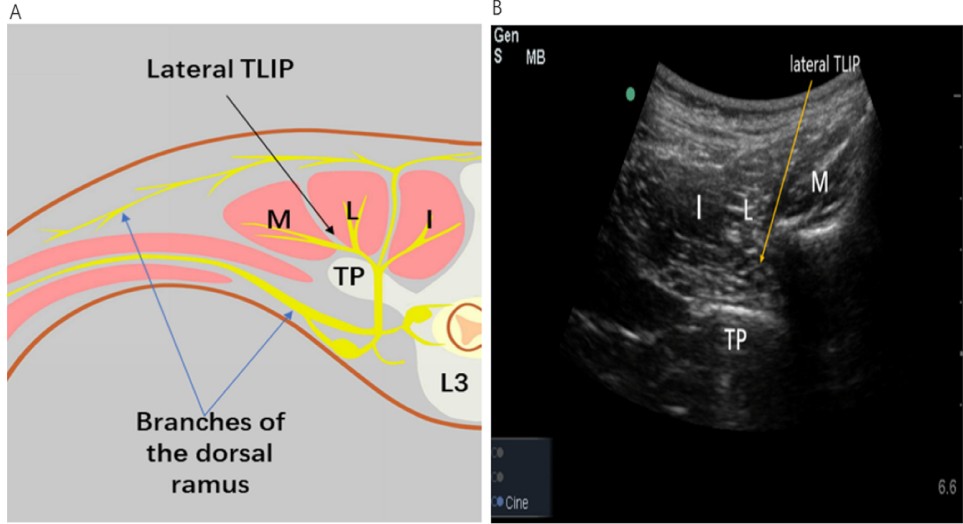

**Figure 2  Image of the spread of lateral TLIP block.** (A) Illustration landmarks and needle approach to the TLIP block. (B) Sonographic image for the TLIP block. The LA injected between the erector spinae muscle and the tip of the transverse processes, anesthetizes the dorsal ramus of the spinal nerves and their branches that innervate the paraspinal muscles and bony vertebrae. L = longissmus muscle; I = illocostalis muscle; M = multifidus muscle; TP = transverse process;SP = spinous process; L3 = lumbar vertebrae 3; lateral TLIP = lateral thoracolumbar Interfascial plane. (Source credit part A: Ke Chen).

in a medial-to-lateral orientation using an ultrasound to guide it through the belly of the longissimus muscle toward the iliocostalis muscle The anesthesiologist injected LA into the interfascial plane between the longissimus muscle (LF) and multifidus muscles (MFs) to anesthetize the dorsal rami of the spinal nerves and relevant branches that innervate the paraspinal muscles and bony vertebrae.

**Table 2  Comparison of anesthesic consummation, PCA and postoperative recovery.**

| Characteristic variable | Group T | Group C | $\chi^2/t/U$ | P value |
|---|---|---|---|---|
| Propofol | 544.03 ± 135.35 | 707.33 ± 191.69 | −3.812 | <0.001 |
| Sufentanil | 26.67 ± 5.31 | 42.50 ± 6.40 | −10.432 | <0.001 |
| PCA compressions | 3.87 ± 0.94 | 6.47 ± 0.90 | −10.963 | <0.001 |
| Remifentanil | 0(0,837.25) | 1,490(1213.50,1657.50) | 63.000 | <0.001 |
| Hospital stay time | 5.53 ± 0.63 | 6.57 ± 0.63 | −6.378 | <0.001 |
| Postoperative days in hospital bed | 11.03 ± 0.72 | 12.93 ± 0.74 | −10.093 | <0.001 |
| Remedial analgesic administration | | | 0.373 | 0.542 |
| No | 24(40.00%) | 22(36.67%) | | |
| Yes | 6(10.00%) | 8(13.33%) | | |

**Table 3  Comparison of postoperative complication.**

| Characteristic variable | Group T | Group C | $\chi^2$ | P value |
|---|---|---|---|---|
| Skin pruritus | | | 0.001 | >0.999 |
| No | 27(45.00%) | 27(45.00%) | | |
| Yes | 3(5.00%) | 3(5.00%) | | |
| Respiratory depression | | | 0.218 | 0.640 |
| No | 28(46.67%) | 27(45.00%) | | |
| Yes | 2(3.33%) | 3(5.00%) | | |
| Sleepiness | | | 0.111 | 0.739 |
| No | 25(41.67%) | 24(40.00%) | | |
| Yes | 5(8.33%) | 6(10.00%) | | |
| Nausea or vomiting | | | 0.111 | 0.739 |
| No | 25(41.67%) | 24(40.00%) | | |
| Yes | 5(8.33%) | 6(10.00%) | | |

## Primary outcome

Perioperative (intraoperative and postoperative) sufentanil (26.67 ± 5.31 vs. 42.50 ± 6.40, $P < 0.01$) and intraoperative remifentanil (0(0,837.25) vs. 1,490(1213.50, 1,657.50), $P < 0.01$) consumption were decreased in group T compared with group C.

## Second outcomes

The frequency of PCA compressions was decreased in Group T compared with group C ($P < 0.05$); however, administration of rescue analgesia was not decreased in Group T compared with Group C ($P > 0.05$). The number of postoperative days in a hospital bed and overall hospital stay time were shorter in Group T compared with Group C (Table 2).

There was no significant difference between the two groups in terms of postoperative complications (Table 3).

The VAS scores at 12, 24 and 36 h postoperatively in Group T were significantly lower than those in Group C; however, there were no significant differences at other times ($P < 0.05$). The BCS scores at 12, 24, 36 and 48 h were significantly lower in Group C compared with Group T (Table 4, Fig. 3).

**Table 4 ANOVA two groups of repeated measurement data at different times.**

| Variation source | VAS(Movement) | | | VAS(Rest) | | | BCS | | |
|---|---|---|---|---|---|---|---|---|---|
| | *df* | *F* | *P* | *df* | *F* | *P* | *df* | *F* | *P* |
| Time | 3.166 | 123.276 | <0.001 | 3.271 | 129.414 | <0.001 | 3.711 | 19.865 | <0.001 |
| Time*group | 3.166 | 36.480 | <0.001 | 3.271 | 86.056 | <0.001 | 3.711 | 2.965 | 0.024 |
| Group | 1 | 184.315 | <0.001 | 1 | 202.790 | <0.001 | 1 | 21.621 | <0.001 |

**Notes.**
Vas, Visual Analogue Scale; BCS, Bruggrmann Comfort Scale.

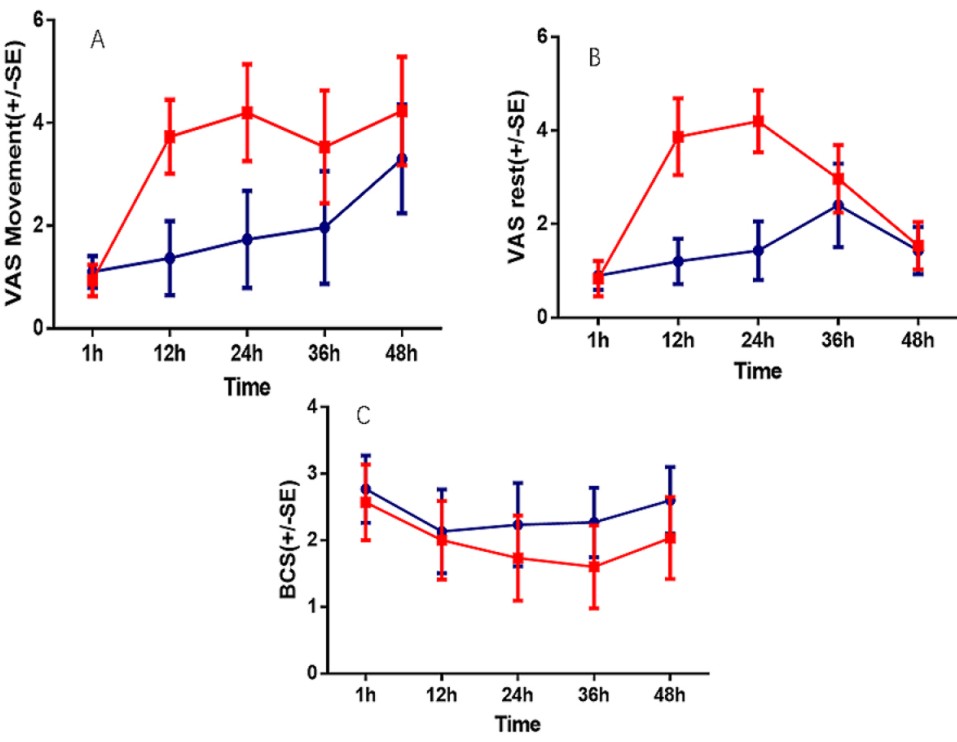

**Figure 3 Comparison of VAS andBCS at post-operation 1 h, 12 h, 24 h, 36 h, 48 h.** A, VAS at movement time; B, VAS at rest time; C, Bruggrmann Comfort Scale.

# DISCUSSION

Our study found that TLIP-block in conjunction with general anesthesia decreased the amount of sufentanil and PCA compressions, significantly decreased the VAS scores and improved postoperative satisfaction in patients undergoing lumbosacral spinal fusion surgery within 48 h following operation.

Posterior lumbar spinal fusion surgery is known to be one of the most painful surgical procedures (*Gerbershagen et al., 2013*). Although the number of spinal surgeries has been increasing for many years, the methods for perioperative pain relief have remained limited. Traditionally, high-dose opioid therapy was used for treating postoperative pain. However, significant side effects and the risk of long-term habituation has limited its use. Reducing

both perioperative opioid requirements and postoperative pain is one of the goals of enhanced recovery after surgery (*Ince et al., 2019*). TLIP block, which blocks the dorsa rami of thoracolubar nerves, has recently been reported as a useful method for preoperative anesthesia in some cases, such as multilevel lumbar spinal surgery (*Ueshima, Oku & Otake, 2016a*; *Ueshima, Sakai & Otake, 2016b*). In our study, the pain score in the TLIP group at rest and movement were lower compared with the control group.

TLIP block provides analgesia depending on the level of the injection site. A study has confirmed the loss of sensory block is 217+84.7 cm$^2$ at 20 min after injection at the lower back[10] (*Hand et al., 2015*). As the dorsal rami of the spinal nerves innervate the paraspinal muscles and posterior bony elements of the spine (*Ince et al., 2019*), the LA may spread to the fascial plane between the MF and LFs of the thoracolumbar spine and exert its analgesic effect via the dorsal rami of the spinal nerves at multiple levels (Fig. 2). Compared with TLIP block, erector spinae plane block (ESPB) inject local anesthetic into a deeper site. Therefore, the treatment for complications, such as hematoma after alternative nerve blocks, may be delayed. In our study, no complications of TLIP block were reported.

In addition, acute postoperative pain relief to conventional multimodal analgesia may effectively prevent the development of postoperative pain syndromes. To discharge patients more quickly after surgery and minimize opioid consumption, multimodal analgesia, including regional blocks, was used to reduce the consumption of other analgesics and their side effects (*Konstantatos et al., 2019*). We found that consumption of opioids decreased in Group T compared with Group C. The postoperative BCS scores of the patients were significantly decreased in our study. We also found that the number of effective PCA compressions in the TLIP group was significantly lower than that in the control group. As a result, perioperative pain scores were reduced and patient satisfaction was improved, enabling patients to both get out of bed and be discharged earlier.

There were some limitations to our study. First, we could not detect the lost sensory area in all enrolled patients after the block procedures because of general anesthesia. We were therefore unsure whether or not there was any regional block failure. However, the anesthesiologist who performed the procedures had 40 to 50 cases of TLIP block experience and could see the local anesthetic spread through the fascial spaces with ultrasound guidance. Second, the study was limited in scope by the lack of wide range applications with increasing popularity, optimal LA volumes and concentrations, complications and adverse effects will soon be reported. Different concentrations and volumes as well as varying types of LAs or mixtures are still topics of research for TLIP. In the future, we will develop further clinical studies, such as radiologic studies, to investigate the proportional relationship between the volume injected and the degree of analgesia.

## CONCLUSION

Our study showed that US-TLIP block could provide sufficient analgesia for lumbar spinal fusion surgery and significantly reduced patient opioid and anesthetic consumption.

Additionally, it could reduce the hospitalization time of patients and improve the satisfaction of patients during the perioperative period.

### Funding

The authors received no funding for this work.

### Competing Interests

The authors declare there are no competing interests.

### Author Contributions

- Ke Chen conceived and designed the experiments, performed the experiments, prepared figures and/or tables, authored or reviewed drafts of the paper, approved the final draft.
- Lizhen Wang and Lianjie Dou analyzed the data, prepared figures and/or tables, approved the final draft.
- Meng Ning and Wei Li performed the experiments, contributed reagents/materials/-analysis tools, authored or reviewed drafts of the paper, approved the final draft.
- Yuanhai Li conceived and designed the experiments, authored or reviewed drafts of the paper, approved the final draft.

### Human Ethics

The following information was supplied relating to ethical approvals (i.e., approving body and any reference numbers):

This research was performed at the First Affiliated Hospital of Anhui Medical University after approval from the Ethics Committee of the First Affiliated Hospital of Anhui Medical University (Ethical Application Ref:2019-01-01MT).

### Clinical Trial Ethics

The following information was supplied relating to ethical approvals (i.e., approving body and any reference numbers):

The research was performed in the first afflicted hospital of Anhui Medical University after Local Ethics Committee approval (Ethical Application Ref:2019-01-01MT).

### Data Availability

The raw measurements are available as a Supplemental File.

### Clinical Trial Registration

The following information was supplied regarding Clinical Trial registration:

ChiCTR1900022233

http://www.chictr.org.cn/showprojen.aspx?proj=37386.

### Supplemental Information

Supplemental information for this article can be found online at http://dx.doi.org/10.7717/peerj.7967#supplemental-information.

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
