# Peer review of "Evaluation of ultrasound-guided lateral thoracolumbar interfascial plane block for postoperative analgesia in lumbar spine fusion surgery: a prospective, randomized, and controlled clinical trial"

_PeerJ, doi:10.7717/peerj.7967_

## Round 0.1 · original submission · Major Revisions

The reviewers have different recommends. Therefore, I invite you to respond to the reviewers' comments in full, and revise your manuscript accordingly.

Reviewer 1 ·

Basic reporting

Your paper would be greatly enhanced by an expert in the field who is fluent in English. There are numerous grammatical errors and some unambiguous text.
Please check the title and body of the article. Is it consistent? For example:
Line 1: lateral thoracolumbar interfascial plane block.
Line 31: bilateral single-shot US-TLIP.

Experimental design

Does saline injection in the control group increase postoperative pain in patients?

Validity of the findings

no comment

Additional comments

Why does nerve block occur after general anesthesia?
How to evaluate the effect of nerve block under general anesthesia in patients?

Reviewer 2 ·

Basic reporting

The language need to be proficiently edited. Not enough information about the research background as well as many repetitions of the same ideas.

Experimental design

Methods are not clear. primary & secondary end points wasn't mentioned.

Validity of the findings

Many assumptions in the discussion weren't supported from the current results or previous studies

Additional comments

- The number of patients enrolled is inconsistent. Why is there difference between number of patients enrolled under the patients & methods vs under the results?

-line 76 the block is for ventral rami of spinal nerves, line 155 the block is for dorsal rami
- A detailed anatomy and sono-anatomy of the area to be blocked should be elaborated
- line 157, you mentioned that the perioperative opioid/propofol requirement are reduced. In a research examining the analgesic efficacy of an intervention, a more detailed information about the exact timing (pre, intra and postoperative), the degree of ….etc opioid requirements.

-Furthermore; no comment whatsoever on whether there’re complications related to opioids, Local anesthetics or the regional technique.

- what kind of spinal surgery enrolled patient underwent?

- please be advised to revise how to write a manuscript discussion. And rewrite the discussion again.

Annotated reviews are not available for download in order to protect the identity of reviewers who chose to remain anonymous.

Reviewer 3 ·

Basic reporting

no comment

Experimental design

no comment

Validity of the findings

no comment

Additional comments

I am glad to see your paper. Your paper was postoperative analgesic efficacy of lateral TLIP block. TLIP block was one of the most attention nerve block to now, I am very interested in the TLIP block.
Although your paper was mentioned the prospective study for the first time, it was wrong (PMID: 29318534). Therefore, it was not novelty. I recommended you to mention clearly why the paper was important.
Best regards.

---

## Round 0.2 · Minor Revisions

This manuscript is an improvement of the previous version. Please add the relevant responses to the comments of the reviewers into the manuscript, such as the reason that nerve block occurs after general anesthesia.

Reviewer 3 ·

Basic reporting

Normal

Experimental design

Normal

Validity of the findings

The dindings was clear. However, it was not novelty.

Additional comments

I'll leave it to you

---

## Round 0.3 · accepted · Accept

This manuscript can be accepted now.